# Rational Design and Characterization of Trispecific Antibodies Targeting the HIV-1 Receptor and Envelope Glycoprotein

**DOI:** 10.3390/vaccines12010019

**Published:** 2023-12-23

**Authors:** Jinhu Liang, Linlin Zhai, Zuxin Liang, Xiaoling Chen, Yushan Jiang, Yuanlong Lin, Shiyan Feng, Yingxia Liu, Wei Zhao, Fuxiang Wang

**Affiliations:** 1Shenzhen Key Laboratory of Pathogen and Immunity, National Clinical Research Center for Infectious Disease, State Key Discipline of Infectious Disease, Shenzhen Third People’s Hospital, Second Hospital Affiliated to Southern University of Science and Technology, Shenzhen 518112, China; 12032630@mail.sustech.edu.cn (J.L.); ganransanke-01@szsy.sustech.edu.cn (Y.L.); ganransanke-02@szsy.sustech.edu.cn (S.F.); yingxialiu@hotmail.com (Y.L.); 2BSL-3 Laboratory (Guangdong), Guangdong Provincial Key Laboratory of Tropical Disease Research, School of Public Health, Department of Laboratory Medicine, Zhujiang Hospital, Southern Medical University, No.1023, South Shatai Road, Baiyun District, Guangzhou 510515, China; zll15649856209@163.com (L.Z.); liangzx2014@outlook.com (Z.L.); wangguosong2020@xmu.edu.cn (X.C.); yushan.y.jiang@gmail.com (Y.J.)

**Keywords:** HIV-1, neutralizing antibodies, trispecific antibodies, host receptor CD4, CCR5

## Abstract

Multitudinous broadly neutralizing antibodies (bNAbs) against HIV-1 have been developed as novel antiviral prophylactic and therapeutic agents. Combinations of bNAbs are generally even more effective than when they are applied individually, showing excellent neutralization coverage and limiting the emergence of escape mutants. In this study, we investigated the design and characterization of three trispecific antibodies that allow a single molecule to interact with independent HIV-1 envelope determinants—(1) the host receptor CD4, (2) the host co-receptor CCR5 and (3) distinct domains in the envelope glycoprotein of HIV-1—using an ELISA, an HIV-1 pseudovirus neutralization assay and in vivo antiviral experiments in humanized mice. We found that trispecific bNAbs and monovalent ones all had satisfactory binding activities against the corresponding antigens in the ELISA, exhibited higher potency and breadth than any previously described single bnAb in the HIV-1 pseudovirus neutralization assay and showed an excellent antiviral effect in vivo. The trispecific antibodies simultaneously recognize the host receptor CD4, host co-receptor CCR5 and HIV-1 envelope glycoprotein, which could mean they have promise as prophylactic and therapeutic agents against HIV-1.

## 1. Introduction

Due to the development of multiple antiviral drugs, HIV-1 has become a treatable disease with near-to-normal life expectancy; however, better tolerated antiviral agents with superb efficacy, few side effects and the ability to limit escape viruses still need to be developed [1]. Anti-HIV-1 drugs cannot only be used for post-viral infection treatment but also the pre-exposure prevention of HIV-1. Currently, the primary pre-exposure prophylactic drugs for HIV-1 that have been marketed are Tenofovir Alafenamide (TAF) [2,3] and Dapivirine [4,5]. However, these drugs may have some side effects, and there are also some HIV-1-resistant strains [6,7]. Developing new and practical pre-exposure prophylactic drugs against HIV-1 can help humans more effectively prevent HIV-1 infection. A variety of broadly neutralizing antibodies (bnAbs) have been generated with broad neutralization spectra and excellent antiviral activities in recent years [8,9,10], but their abilities to treat HIV-1 patients may be limited by the breadth and potency of their viral neutralization [11,12]. HIV-1 bnAbs have been characterized to recognize several conserved regions on the HIV-1 envelope glycoprotein (Env); these include the CD4 binding site (e.g., N6 [13]), the N332 glycan supersite (e.g., PGT121) [14], the membrane-proximal external region of gp41 (e.g., 10E8 [15]) and the V2 apex (e.g., PGDM1400) [16]. In addition, the host receptor and co-receptor of HIV-1 are also important targets for bnAbs because the receptor proteins do not mutate as a virus does—the antibodies that target the receptor possess extraordinary neutralization breadth and efficacy against diverse HIV-1 strains and inhibit the virus from entering host cells [17].

iMab, an anti-CD4 antibody, has shown a broad spectrum of antiviral effects and prevents HIV-1 infection in a non-competitive manner [18]; another antibody, PRO140, exerts effective antiviral potency by inhibiting the binding between HIV-1 and the host co-receptor CCR5 [19]. The HIV-1 virus can generate escape mutants rapidly under pressure from a single bNAb; while antibody cocktails have shown improved antiviral efficacy and breadth in preclinical studies [20], multispecific “single agents” are desirable for enhancing neutralization breadth and potency [17,21,22]. bNAbs, which recognize the CD4 binding site, variable region glycans and MPER, have been engineered into various multispecific antibodies and provide optimal neutralization [22,23]. However, to date, no trispecific antibodies that simultaneously recognize the host CD4 receptor, CCR5 receptor and HIV-1 envelope glycoprotein have been developed; this study aims to explore the antiviral activity and breadth of this class of trispecific antibodies to provide a supplement for the development of novel antiviral drugs for HIV-1.

## 2. Materials and Methods

### 2.1. Expression of Recombinant Human CD4 and CCR5 Proteins

Recombinant CD4 protein (aa 26-396, detailed sequences are provided in the Appendix A) and CCR5 protein (aa 1-352, detailed sequences are provided in the Appendix A) were expressed in human HEK293F cells as soluble proteins. The soluble human CD4 and CCR5 proteins were purified using a HisTrap HP column (GE Healthcare, Chicago, United States) and were further purified via size-exclusion chromatography with a Superdex 200 column (GE Healthcare). Then, the purified proteins were dialyzed in 1×PBS dialysate, and the dialysate was replaced at 4 h intervals. Finally, the dialyzed protein was concentrated to 1 mg/mL at 4 °C with an ultrafiltration tube, and samples were analyzed via SDS-PAGE and stored at −80 °C until use.

### 2.2. Construction, Expression, and Purification of IgG and Trispecific Antibodies

iMab, PRO140, 10E8, PGT121 and PGDM1400 were cloned as human IgG1 antibodies in cytomegalovirus (CMV)-driven expression plasmids [24]. Trispecific molecules were engineered in the DVD-Ig format, and the sequences for two scFvs were cloned in frame with sequences encoding connecting G4S linkers on both the N and C termini of the full IgG1 antibody. Specifically, the iMab variable domain of the heavy chain and PRO140 variable domain of the heavy chain were fused with a GGGGSGGGGS linker, followed by a constant region (CH1-CH2-CH3); then, each ScFv from 10E8, PGDM1400 and PGT121 was connected to the C terminus of the CH3 via a GGGGSGGGGS linker in the heavy chain. For the light chain of the trispecific antibodies, the iMab variable domain and PRO140 variable domain were connected via the GGGGSGGGGS linker, followed by a constant region (CL). The plasmids encoding the heavy chain and light chain were cotransfected into HEK293F cells with a 1:1.5 molar ratio of the heavy chain and the light chain. Cells were cultured with SMM 293-TII medium (Sinobiological, Beijing, China) at 310 K and 5% CO_2_; then, the supernatants were collected on the fifth day, filtered through a 0.22 μm membrane, purified via protein A and then further purified via size-exclusion chromatography in PBS buffer (pH 7.4). The small-scale production of transfected HEK293F cell culture of the three trispecific antibodies iMab+PRO140+10E8, iMab + PRO140 + PGMD1400 and iMab + PRO140 + PGT121 is 5–10 mg/liter. The associated antibody amino acid sequences are presented in the Appendix A.

### 2.3. ELISA

In the next step, 96-well microtiter plates were coated with 100 ng/well-purified proteins or viruses overnight at 4 °C. The plates were washed thrice with PBST (1 × PBS containing 0.1% *v*/*v* Tween-20) and blocked with 200 μL blocking buffer (1 × PBS containing 2% *w*/*v* non-fat dry milk) at 37 °C for 2 h. The microtiter plates were then washed once with 1*PBST and shaken dry. Serial dilutions of purified HIV-1 neutralizing monoclonal antibodies were added to the wells at 100 μL/well and incubated for 30 min at 37 °C. Following that, 100 μL of horseradish peroxidase (HRP)-conjugated goat anti-human IgG antibody solution was added to each well after three washes, and incubated at 37 °C for 30 min. Then, 100 μL of tetramethylbenzidine (TMB) substrate was added to each well after five washes, at room temperature in the dark. The reaction was stopped with a 2 M H_2_SO_4_ solution after 15 min, and the absorbance was measured at 450 nm. All samples were run in triplicate. The relative affinity of antibody binding to purified viruses or proteins was determined by measuring the concentration of antibody required to achieve the EC_50_. 

### 2.4. K_d_ Determination

HIV-1 proteins were amine coupled to a carboxylated dextran polymer matrix, which was coated on CM-5 sensor chips for use in the BIAcore 3000. The carboxyl groups on the chips were activated with 35 µL of a 1:1 (*v*/*v*) solution of N-hydroxysuccinimide and N-ethyl-N-(3-diethylaminopropyl) carbodiimide for 7 min. HIV-1 proteins at a concentration of 50 µg/mL were prepared using 20 mM sodium acetate as a diluent and each coated onto one cell, whereas the other cell was left blank as a reference and blocked with 1 M ethanolamine HCl, pH 8.5 (GE Healthcare, Chicago, United States) as a control. The affinity measurements for antibody binding with different antigens were started by allowing HBS (10 mM HEPES (N-2-hydroxyethylpiperazine-N’-2-ethanesulfonic acid), 150 mMNaCl and 3 mM EDTA) to flow over the sensor surface at 30 µL/minute for 100 s of baseline acquisition, followed by injection of each serially diluted antibody at 30 µL/minute for approximately 100 s. The results were used to fine-tune the affinity fit. The ratio of k_on_ to k_off_ determined the K_d_ reported here.

### 2.5. Virus Neutralization Assay

HIV-1 viral neutralization by the neutralizing antibodies was assessed using a single-cycle assay with huCD4^+^ CCR5^+^TZM-BL target cells and HIV-1 pseudoviruses, as described previously [21]. The parental antibodies or the trispecific antibodies were serially diluted and incubated with the HIV-1 pseudoviruses to determine the IC_50_ values of each antibody, which reflect the neutralization of HIV-1. The IC_50_ values were determined using the GraphPad Prism 6.0 software by fitting the neutralization dose–response curves using nonlinear regression. 

### 2.6. Humanized Mouse Model

Animal procedures that might cause more than slight pain or distress were performed with appropriate anesthesia. Immunodeficient NOG mice were purchased from Beijing Vital River Laboratory Animal Technology Co., Ltd (Beijing, China). The 4- to 6-week-old NOG mice were humanized as previously described [25].

### 2.7. HIV-1 RNA Load Measurement

HIV-1 RNA was extracted using the QIAamp viral RNA mini kit and reverse transcribed to cDNA with the RT-PCR Prime Script Kit from Takara. Then, 2 μL cDNA was used in a 20 μL qRT-PCR reaction using the Takara PCR Master Mix with the specific HIV-1 P17 gene primers (5′-TCTCGA CGCAGG ACTCG-3′ and 5′-TACTGA CGCTCT CGCACC-3′) and a TaqMan probe (5′-FAM-CTCTCT CCTTCT AGCCTC-MGB-3′), in the following conditions: 1 cycle of 50 °C for 2 min, 1 cycle of 95 °C for 10 min and 40 cycles of 95 °C for 15 s and 60 °C for 1 min. The HIV-1 viral titer was determined via comparison with a standard curve generated using RNA extracted from a serially diluted reference HIV-1 viral stock. 

### 2.8. In Vivo Assay in Humanized Mice

All animal experimental procedures were approved by the Committee on the Use of Live Animals in Shenzhen Third People’s Hospital. The t_1/2_ of the three trispecific antibodies was computed as ln (2)/k, where k is a rate constant expressed reciprocally of the x-axis time units by the 1-phase decay equation in the GraphPad software (version 6.0). One day before the HIV-1 challenge, 200 μg (10 mg/kg) or 20 μg (1 mg/kg) of each antibody was injected intraperitoneally (i.p.) into NOG mice; after 24 h of antibody administration, the mice were challenged through the i.p. route with 500 TCID_50_ live HIV-Shenzhen-_BC_. These mice were subjected to daily blood sampling to monitor viral load for 14 days. 

### 2.9. Antibody-Dependent Cell-Mediated Cytotoxicity (ADCC) Assay

ADCC was determined using a previously described assay utilizing two fluorescent dyes to distinguish between viable and dead cells [24]. TZM-BL cells were labeled with a general cell membrane labeling dye PKH67 (Sigma-Aldrich, Darmstadt, Germany). The cells were then washed twice with PBS, added to 96-well plates in 5 × 10^4^ cells per well in triplicate and incubated with serial dilutions of single antibodies (0.5–20 mg/μL) for 30 min at 37 °C. NK cells (natural killer cells) isolated from human PBMCs were added to the plates (1 × 10^6^ per well), incubated at 37 °C for 3 h and treated with 1 μL of 7-AAD (eBioscience, California, United States) to stain dead cells. Cell death was determined using a FACSAria III flow cytometer with BD FACS Diva software (version 6.1.2).

## 3. Results

### 3.1. Determination of Antigen-Binding Activity of the Antibodies Generated in This Study

The monoclonal antibodies and their recognition targets involved in this study are shown in Appendix A. Through HEK293F expression and Protein A purification, we obtained highly purified monoclonal antibodies iMab, PRO140, 10E8, PGT121, PGDM1400 and VRC01; trispecific antibodies iMab + PRO140 + 10E8, iMab + PRO140 + PGMD1400 and iMab + PRO140 + PGT121 were expressed and purified (Appendix A). First, the binding activity of the expressed antibodies against the corresponding antigens was evaluated through the ELISA experiment. CD4 receptor protein, CCR5 co-receptor protein, iMab and PRO140 expressed by eukaryotic cells showed good activity. The EC_50_ value of iMab for the CD4 receptor protein was 17 ng/mL (Figure 1A), and the EC_50_ value of PRO140 for the CCR5 co-receptor was 92 ng/mL (Figure 1B). The EC_50_ values of the 10E8 antibody against gp41 protein of HIV-1 subtype A, B, C and AC strains were 1145, 381, 306 and 434 ng/mL, respectively (Figure 1C). PGDM1400 and PGT121 showed better binding activity to HIV-1 antigen than that of 10E8. The EC_50_ values of PGDM for HIV-1 A, B, C and AC subtypes were 33, 64, 23 and 294 ng/mL, respectively (Figure 1D), while the corresponding values of PGT121 were 57, 69, 115 and 51 ng/mL, respectively (Figure 1E). After characterization of the activity of a single antibody, the antibodies were engineered into three trispecific antibodies in the form shown in Figure 1F: iMab + PRO140 + 10E8, iMab + PRO140 + PGMD1400 and iMab + PRO140 + PGT121, respectively. The three trispecific antibodies maintained favorable binding activity to CD4 receptor and CCR5 co-receptor and showed good binding activity to different subtypes of HIV-1 antigens (Figure 1G–I). In addition, the binding of trispecific antibody iMab + PRO140 + 10E8 to HIV-1 gp41 protein, as well as of trispecific antibodies iMab + PRO140 + PGMD1400 and iMab + PRO140 + PGT121 to HIV-1 gp120 protein, was measured via surface plasmon resonance, with the HIV-1 proteins being firmly bound by the trispecific antibodies: K_d_ (dissociation constant) = 6.12 nM for iMab + PRO140 + 10E8 versus HIV-1 gp41 protein; K_d_ = 0.46 nM for iMab + PRO140 + PGMD1400 versus HIV-1 gp120; K_d_ = 0.59 nM for iMab + PRO140 + PGT121 versus gp120 (Appendix A).

### 3.2. Determination of In Vitro Neutralization Activity against HIV-1 Pseudoviruses and In Vivo Antiviral Activity of the Antibodies Generated in This Study

In further experiments, the neutralizing activities of the monoclonal antibodies, antibody cocktails and three trispecific antibodies against different subtypes of HIV-1 pseudovirus were evaluated. A total of 23 pseudoviruses of HIV-1 were selected in the neutralization assay in this study, including HIV-1 subtype A, subtype B, subtype C, subtype G, subtype AE, subtype AG, subtype BC and subtype AC. All five single monoclonal antibodies showed broad-spectrum and potent activity in neutralizing the HIV-1 virus. However, several sporadic virus strains can escape neutralization by a single antibody. iMab and 10E8 could not neutralize the X1632 strain from HIV-1 subtype G, while KER2008.12 from subtype A and T26-60 from subtype AG escaped the neutralization from 10E8. PRO140 was unable to neutralize the BJOX2000 and XJN0181ENV04 strains from subtype BC, PGDM1400 failed to effectively neutralize one subtype B and one subtype C HIV-1 strain, and PGT121 and the control antibody VRC01 did not show neutralizing activity against four of the HIV-1 strains (Figure 2). In contrast, the three-antibody cocktails showed a more robust and broader spectrum of virus-neutralizing activity than a single antibody. The three cocktails neutralized all viruses, and the trispecific form of antibodies showed the best antiviral activity and reactive spectrum (Figure 2). 

The median IC_50_ neutralizing titers of iMab, PRO140, 10E8, PGMD1400 and PGT121 against all HIV-1 viruses were 1.38, 3.34, 3.18, 2.06 and 3.78 μg/mL, respectively; the antiviral mean IC_50_ titers of the iMab + PRO140 + 10E8, iMab + PRO140 + PGMD1400 and iMab + PRO140 + PGT121 cocktails against all HIV-1 viruses were 0.24, 0.15 and 0.18 μg/mL, while the values of the trispecific antibodies iMab + PRO140 + 10E8, iMab + PRO140 + PGMD1400 and iMab + PRO140 + PGT121 were 0.12, 0.05 and 0.08 μg/mL, respectively (Figure 3). The IC_50_ values of trispecific antibodies iMab + PRO140 + 10E8, iMab+PRO140+PGMD1400 and iMab + PRO140 + PGT121 against the HIV-1 strains showed statistically significant differences compared to the IC_50_ values of monoclonal antibodies iMab, PRO140, 10E8, PGMD1400 and PGT121; in contrast, the IC_50_ values of the three cocktail antibodies showed significant differences compared to the values of iMab, PRO140, 10E8 and PGMD1400, while they did not show significant increases when compared to that of PGT121; the IC_50_ values of the trispecific antibodies iMab + PRO140 + PGDM1400 and iMab + PRO140 + PGT121 were significantly higher than those values of their corresponding parental antibody cocktail combinations (Appendix A).

The dual specificity against CD4 and CCR5 could lead to CD^4+^ T cell depletion in vivo due to several mechanisms such as ADCC; the trispecific antibodies could bind CD4^+^ CCR5^+^ T cells at low concentrations and then recruit effector cells such as NK cells that express the Fc receptor to kill the CD4+CCR5+ T cells. To address this concern, we performed in vitro assays to test the ADCC effect of the trispecific antibodies on TZM-BL cells as target cells and human NK cells as effector cells in the presence of trispecific antibodies. The results showed that the trispecific antibodies iMab + PRO140 + 10E8, iMab + PRO140 + PGMD1400 and iMab + PRO140 + PGT121 showed no ADCC killing activity on TZM-BL cells (Appendix A). We then measured the pharmacokinetics and bioavailabilities of iMab + PRO140 + 10E8, iMab + PRO140 + PGMD1400 and iMab + PRO140 + PGT121 in five healthy NSG-HuPBL mice (2 male (M) and 2 female (F)) and found that the calculated peripheral half-lives (t_1/2_) of iMab + PRO140 + 10E8, iMab + PRO140 + PGMD1400 and iMab + PRO140 + PGT121 were 5.279, 4.355 and 4.378 days, respectively (Figure 4A). These results showed that trispecific iMab + PRO140 + 10E8, iMab + PRO140 + PGMD1400 and iMab + PRO140 + PGT121 were stable in vivo. Finally, the in vivo protective activity of three trispecific antibodies was tested in NOG humanized mice, which were given the antibodies one day before HIV-1 infection. The results showed that HIV titers in the plasma of mice in the antibody-negative control group gradually increased during the two weeks of HIV-1 infection; in contrast, when the antibody was administered at a dose of 10 mg/kg, no HIV-1 RNA was detected in the plasma of mice treated with the three trispecific antibodies (Figure 4B). When the antibody was administered at 1 mg/kg, the HIV-1 virus could be detected on the third day after the infection of the mice in the iMab + PRO140 + 10E8-treated group, and the virus titer gradually increased during the observation period, but the virus titer of the mice treated by the iMAB + PRO140 + 10E8 group was significantly reduced compared with the negative antibody control group. Encouragingly, no HIV-1 virus was detected in mice treated with 1 mg/kg of iMab + PRO140 + PGMD1400 or iMab + PRO140 + PGT121 (Figure 4C).

## 4. Discussion

In recent years, several studies have revealed that broadly neutralizing antibodies against HIV-1 are a class of preventive antiviral drugs with advantages of high efficacy and few side effects, and they have shown promising antiviral effects in laboratory and clinical studies [26,27,28,29], indicating that these may be used as a supplement to the current antiretroviral HIV-1 drugs. A large number of HIV-1 bNAbs have been developed, including bNAbs that recognize various conserved epitopes on HIV-1 envelope glycoprotein, bNAbs that recognize HIV-1 cell receptors and co-receptors, as well as bispecific and trispecific antibodies engineered based on the above single antibodies [30,31,32]. One of the main reasons why HIV-1 can pose a significant challenge to humans is that HIV-1 is very easy to mutate and has many subtypes. However, no matter how the HIV-1 virus mutates, the virus needs to infect cells through cell receptors; therefore, antibodies that competitively bind cell receptors or co-receptors with HIV-1 are up-and-coming antiviral drugs. Examples include anti-CD4 binding site antibodies and CCR5 binding site antibodies [33,34]. Multispecific antibodies can inhibit viral binding to cell receptors by competitively binding to cell receptors or co-receptors and simultaneously inhibit viral replication by binding to important functional epitopes of the HIV-1 virus, thus having dual advantages. Theoretically, this kind of multispecific antibody has dual advantages, possesses stronger antiviral efficacy and is less likely to be escaped by the virus. However, until now, no trispecific antibodies have been reported that simultaneously target the HIV-1 CD4 receptor, CCR5 co-receptor of the host and the functional epitopes of the virus, and this study supplements the current research gap. We selected iMab, PRO140, 10E8, PGT121 and PGDM1400 monoclonal antibodies as parental antibodies to conduct trispecific antibody studies. iMab was developed by Tanox Inc (under license from Biogen Idec, formerly Biogen Inc. Delaware, United States) around the year 2000 [35], PRO-140 was developed by Progenics Pharmaceuticals, Inc. in the 1990s [36], 10E8 was developed by the Mark Connors team at the National Institutes of Health in 2012 [15], PGT121 was developed by Julie Overbaugh’s team from the Fred Hutchinson Cancer Research Center in the United States in 2012 [37] and PGDM1400 was developed by Dennis R Burton’s team from the Massachusetts Institute of Technology and Harvard University in 2014 [16]. We found that the trispecific antibodies simultaneously recognized the CD4 binding site, CCR5 co-receptor and the envelope glycoprotein, meaning they have a better neutralization breadth and potency than a single antibody component. This class of antibodies is expected to be further developed into novel therapeutic and preventive agents against HIV-1. Among the three trispecific antibodies in this study, iMab + PRO140 + PGDM1400 showed the best overall neutralizing activity against HIV-1 and demonstrated decent in vivo anti-HIV-1 activity in humanized mouse models. Therefore, in further research, our team will choose the trispecific antibody iMab + PRO140 + PGDM1400 to conduct further in vivo assays to determine the antiviral efficacy of this antibody in non-human primates. The neutralizing activity of the trispecific antibodies iMab+PRO140+10E8, iMab+PRO140+PGMD1400 and iMab + PRO140 + PGT121 against the HIV-1 virus strains was improved compared to the combination of the three parent antibody cocktails. The neutralizing activity of the trispecific antibodies against most tested HIV-1 strains was superior to the corresponding antibody cocktails, demonstrating the advantages of the trispecific forms of antibodies. At the same time, as single molecules, the trispecific antibodies are more accessible to prepare than antibody cocktails.

In a previous study, it was found that the combination of iMab and 10E8 bispecific antibody could neutralize almost all the HIV-1 pseudoviruses tested, indicating that the combination of the antibody targeting the cellular receptor and an antibody targeting the viral epitopes could have a synergistic antiviral effect and a broad antiviral spectrum [38]. In another previous study, antibodies recognizing different HIV-1 epitopes such as PG9, PG16, PGT128, VRC01 and Hu5A8 were utilized to synthesize bispecific antibodies and it was found that those antibodies had excellent antiviral activity in vitro and in vivo [39]. In this study, we also found that the molecules in the form of trispecific antibodies had stronger antiviral activity and neutralization profiles than the parent antibody. Based on the previous studies, we further introduced anti-CCR5 monoclonal antibodies, a design that could further broaden the neutralization spectrum of multispecific antibodies. The trispecific antibody designed in this study did not show ADCC killing activity on CD4+CCR5+ cells, indicating that these trispecific antibodies have good safety. The three trispecific antibodies developed in this study have high small-scale expression efficiency in HEK293F cells, indicating that their production costs can be effectively controlled and they may have good affordability as marketed drugs in the future.

The novel trispecific antibodies against HIV-1 developed in this study will address some of the limitations of the WHO-recommended pre-exposure prophylactic drugs against HIV-1 that are already on the market for the following reasons [12,28,40]: (1) The trispecific-antibody drugs possess a lower risk of drug resistance: in contrast to trispecific antibodies, certain drugs may pose a risk of pathogen development of tolerance to the medication. (2) Trispecific antibodies exhibit an extended half-life, allowing for sustained immune protection over a defined period. (3) Trispecific antibodies target three distinct antigens of HIV; in contrast, certain traditional antiviral drugs may lack the same specificity, resulting in a lesser impact on host cells. (4) Trispecific antibodies have demonstrated the ability to reactivate and eliminate latently infected cells under prolonged suppression by antiretroviral therapy (ART).

This study also has some limitations: the HIV-1 pseudovirus strains we tested were limited in variety and number, the number of experimental animals tested was limited and experiments were not performed on non-human primates, nor did carry out related clinical experiments, and the types of bnAbs selected did not fully cover all the antibody types generated so far.

## Figures and Tables

**Figure 1 vaccines-12-00019-f001:**
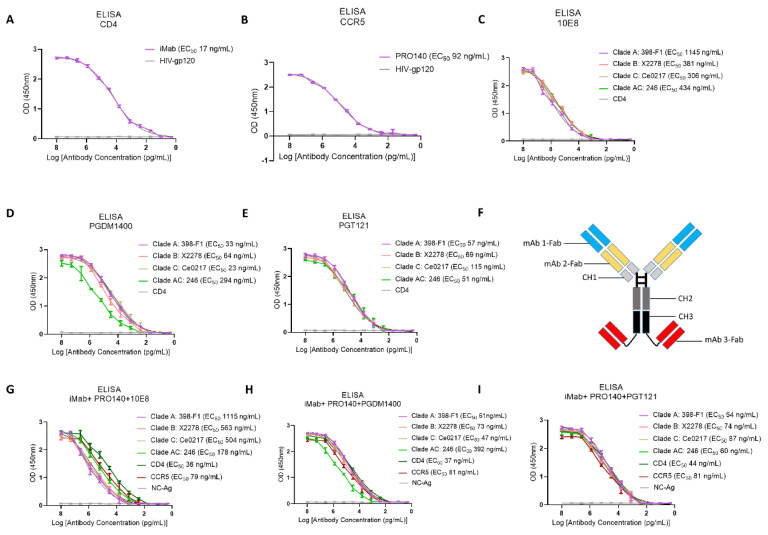
Binding activity of single antibody and trispecific antibody. (**A**) ELISA binding activity of iMab against CD4 receptor. (**B**) ELISA binding activity of PRO140 against CCR5 co-receptor. (**C**–**E**) ELISA binding activity analysis of 10E8 (**C**), PGMD1400 (**D**) and PGT121 (**E**) against different subtypes of HIV-1 envelope glycoprotein. (**F**) Illustration of the design of the trispecific antibody in this study. Blue: variable region of monoclonal antibody 1, yellow: variable region of monoclonal antibody 2, red: variable region of monoclonal antibody 3, light gray: human IgG1 CH1, dark gray: human IgG1 CH2, black: human IgG1 CH3. (**G**–**I**) Neutralizing activity analysis of trispecific antibodies (**G**) iMab + PRO140 + 10E8, (**H**) iMab + PRO140 + PGMD1400 and (**I**) iMab + PRO140 + PGT121 against different HIV-1 pseudoviruses. Each experiment is repeated three times, with double wells for each experiment.

**Figure 2 vaccines-12-00019-f002:**
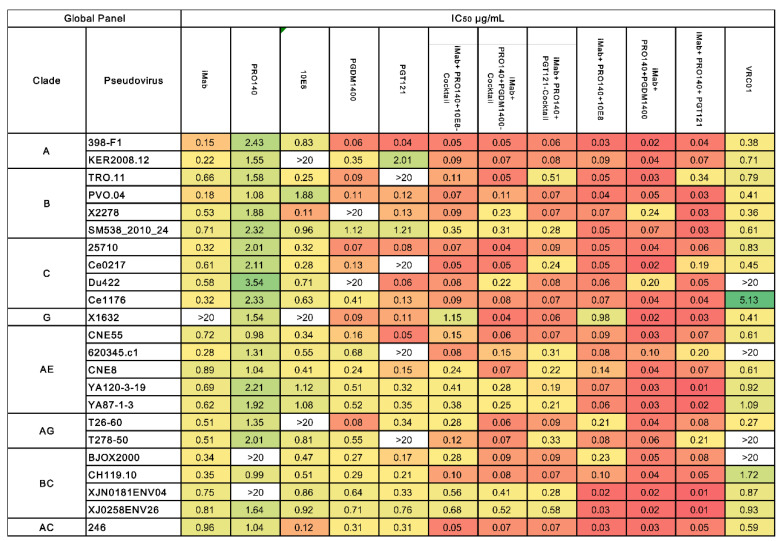
The neutralization breadth of the parental and trispecific antibodies was tested against an HIV-1 pseudovirus panel consisting of Envs of 23 viral strains. Heat maps of IC_50_ titers were generated in GraphPad Prism. In the heatmaps, each row represents a virus strain while columns represent antibodies. Red colors indicate more potent neutralization, and green indicates lower potency; an antibody virus neutralization titer more significant than 20 µg/mL is below the detection threshold. The median IC_50_ value for each antibody is calculated and placed in the last row.

**Figure 3 vaccines-12-00019-f003:**
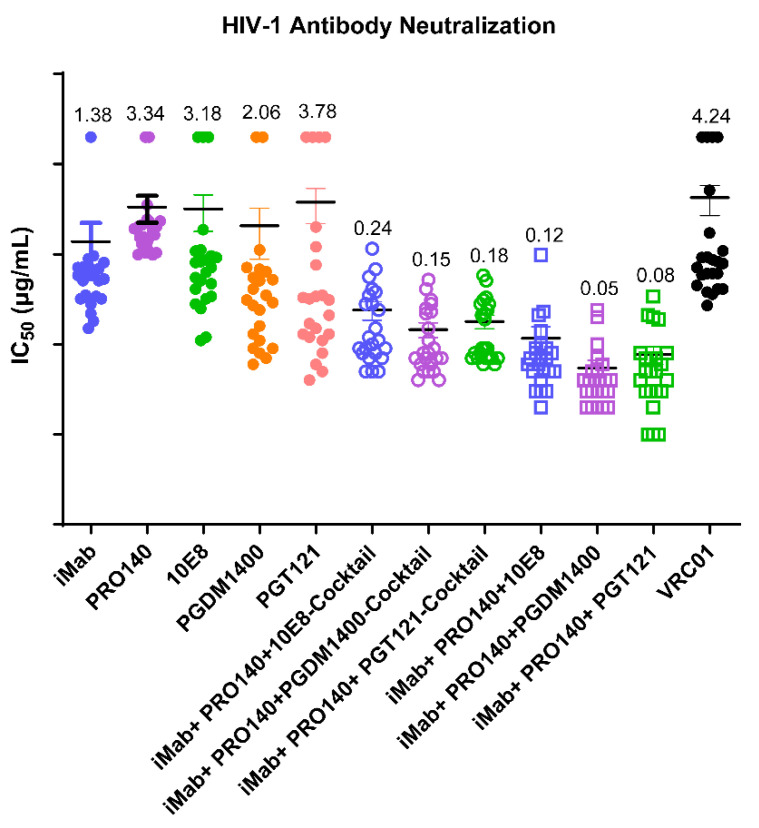
Analysis of IC_50_ values for each antibody from three independent neutralization experiments. IC_50_ values for each antibody against each virus are plotted as symbols. The statistical differences among the groups are shown in Appendix A. IC_50_ values less than 20 are represented by the experimentally obtained IC50 value; IC_50_ values greater than 20 are uniformly represented by 20.

**Figure 4 vaccines-12-00019-f004:**
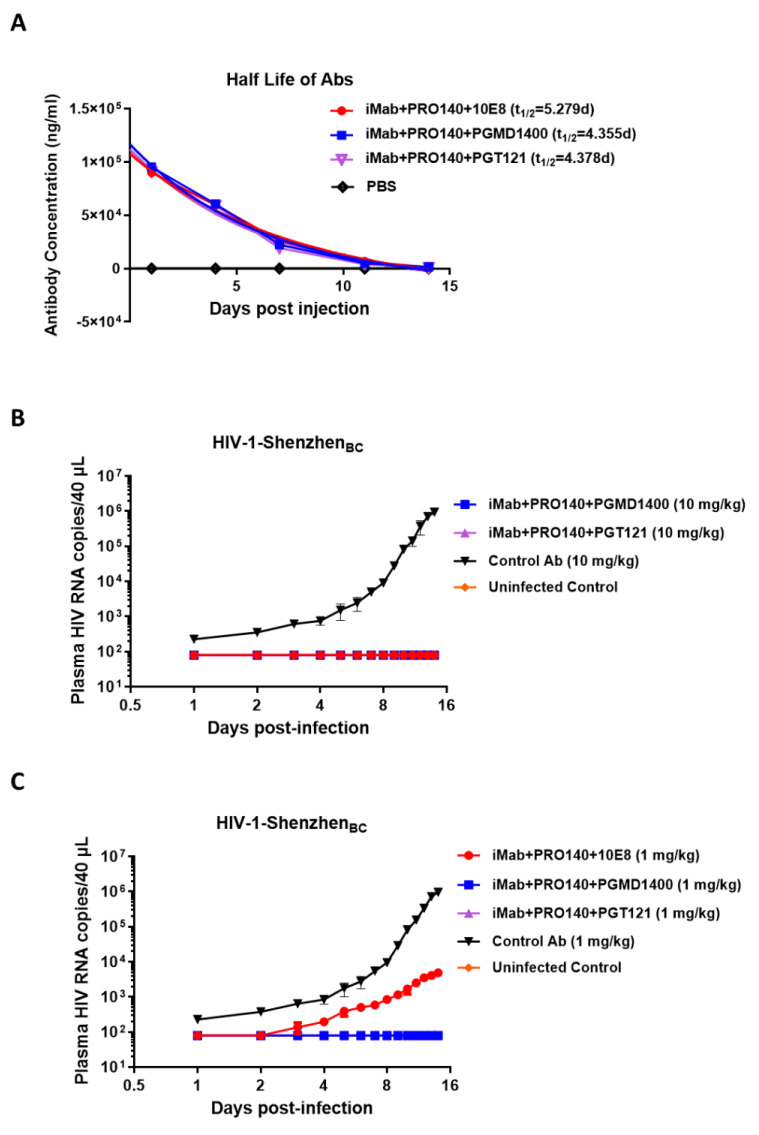
In vivo antiviral activity of HIV-1 trispecific antibodies. (**A**) Bioavailability and t_1/2_ of iMab + PRO140 + 10E8, iMab + PRO140 + PGMD1400 and iMab + PRO140 + PGT121 in NSG-HuPBL mice. Data represent mean ± SEM (*n* = 5). (**B**,**C**) Plasma viral loads among 5 groups of NOG mice including the iMab + PRO140 + 10E8-treated group (red, *n* = 4), iMab + PRO140 + PGMD1400-treated group (blue, n = 4), iMab + PRO140 + PGT121-treated group (violet, *n* = 4) and control Ab-treated group (black, *n* = 4) in (**B**) 10 mg/kg antibody or (**C**) 1mg/kg antibody and the uninfected control (orange, *n* = 4), where the mice were challenged with the HIV-1-Shenzhen BC subtype strain. ADCC of iMab + PRO140 + 10E8, iMab + PRO140 + PGMD1400 and iMab + PRO140 + PGMD1400.

## Data Availability

The raw data supporting the conclusions of this article will be made available by the authors, without undue reservation.

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
