# Peer review of "Rational Design and Characterization of Trispecific Antibodies Targeting the HIV-1 Receptor and Envelope Glycoprotein"

_vaccines, 2023, doi:10.3390/vaccines12010019_

Round 1

Reviewer 1 Report

Comments and Suggestions for Authors

Liang et al. presented a study on the construction and characterization of three trispecific antibodies, each targeting the HIV-1 receptor CD4, co-receptor CCR5, and the HIV-1 envelope glycoprotein. The authors conducted evaluations on their in vitro binding and neutralizing capabilities, along with an assessment of their in vivo prophylactic effects. While the study claims that these trispecific antibodies surpass the parental antibody combinations in terms of neutralization potency and breadth, the provided data do not convincingly support this conclusion.

Major Concerns:

  1. A critical aspect of these synthetically engineered trispecific antibodies is their in vivo uptake and sustained efficacy. However, the manuscript lacks essential data on the stability and in vivo kinetics of these antibodies, which is a significant omission.
  2. Neutralization data from a global HIV-1 clade panel shows that the trispecific antibodies do not demonstrate superior median neutralizing IC50 titers, compared to the combined parental antibodies (0.16 vs 0.14, 0.08 vs 0.06, and 0.15 vs 0.10, respectively). This raises questions about the claimed effectiveness of the trispecific antibodies and impacts the overall significance of the findings.

Minor Points:

  1. The study requires more statistical analyses, particularly comparing the IC50 values of the trispecific antibodies against the combined parental antibodies.
  2. It is necessary to define 'i.p.' and clarify the time interval between antibody administration and the subsequent virus challenge (lines 130-132 on page 3).
  3. The color coding in Figure 1f should be clearly defined for better understanding.
  4. Certain result descriptions are inaccurate. For instance, the data in Figure 2 indicate that VRC01 failed to neutralize 4 out of 18 pseudoviruses, not 2 as stated.

Comments on the Quality of English Language

The manuscript requires language refinement for clarity and consistency in tense usage.

Author Response

Review 1: Liang et al. presented a study on the construction and characterization of three trispecific antibodies, each targeting the HIV-1 receptor CD4, co-receptor CCR5, and the HIV-1 envelope glycoprotein. The authors conducted evaluations on their in vitro binding and neutralizing capabilities, along with an assessment of their in vivo prophylactic effects. While the study claims that these trispecific antibodies surpass the parental antibody combinations in terms of neutralization potency and breadth, the provided data do not convincingly support this conclusion.

Major Concerns:

  1. A critical aspect of these synthetically engineered trispecific antibodies is their in vivo uptake and sustained efficacy. However, the manuscript lacks essential data on the stability and in vivo kinetics of these antibodies, which is a significant omission.

Response: Thank you very much for your suggestion. We have added data on the in vivo stability of three specific antibodies. The t1/2 of the three trispecific antibodies was computed as ln (2)/k, where k is a rate constant expressed reciprocally of the x axis time units by the 1-phase decay equation in the GraphPad software. We measured the pharmacokinetics and bioavailabilities of iMab+PRO140+10E8, iMab+PRO140+PGMD1400 and iMab+PRO140+PGT121 in 5 healthy NSG-HuPBL mice (2 male [M] and 2 female [F]) and found that the calculated peripheral half-life (t1/2) of iMab+PRO140+10E8, iMab+PRO140+PGMD1400 and iMab+PRO140+PGT121 were 5.279, 4.355 and 4.378 days, respectively (Figure 4A). These results showed that trispecific iMab+PRO140+10E8, iMab+PRO140+PGMD1400 and iMab+PRO140+PGT121 were stable in vivo.

Please refer to lines 172-174, lines 296-303 and Figure 4A.

  1. Neutralization data from a global HIV-1 clade panel shows that the trispecific antibodies do not demonstrate superior median neutralizing IC50 titers, compared to the combined parental antibodies (0.16 vs 0.14, 0.08 vs 0.06, and 0.15 vs 0.10, respectively). This raises questions about the claimed effectiveness of the trispecific antibodies and impacts the overall significance of the findings.

Response: We have added 5 strains of HIV-1 pseudovirus isolated from China to the virus panel. The new data show that the antiviral mean IC50 titers of the iMab+PRO140+10E8, iMab+PRO140+PGMD1400 and iMab+PRO140+PGT121 cocktails against all HIV-1 viruses were 0.24, 0.15 and 0.18 μg/mL, while the values of the trispecific antibodies iMab+PRO140+10E8, iMab+PRO140+PGMD1400 and iMab+PRO140+PGT121 were 0.12, 0.05 and 0.08 μg/mL, respectively (Figure 3). The IC50 values of the trispecific antibodies iMab+PRO140+PGDM1400 and iMab+PRO140+PGT121 were significantly higher than those values of their corresponding parental antibody cocktail combinations (Table S2).

Please refer to lines 267-280.

The neutralizing activity of the tri-specific antibodies iMab+PRO140+10E8, iMab+PRO140+PGMD1400, and iMab+PRO140+PGT121 against the HIV-1 virus strains has been improved compared to the combination of the three parent antibody cocktails. The neutralizing activity of the trispecific antibodies against most tested HIV-1 strains is superior to the corresponding antibody cocktails, demonstrating the advantages of the tri-specific forms of antibodies, at the same time, as a single molecule, the tri-specific antibodies are more accessible to prepare than antibody cocktails.

Please refer to lines 368-375.

Minor Points:

  1. The study requires more statistical analyses, particularly comparing the IC50 values of the trispecific antibodies against the combined parental antibodies.

Response: We have added a new statistical analysis table as per your suggestion, please refer to Table S2 and lines 271-280.

  1. It is necessary to define 'i.p.' and clarify the time interval between antibody administration and the subsequent virus challenge (lines 130-132 on page 3).

Response: We have added the information, one day before the HIV-1 challenge, 200 μg (10 mg/kg) 20 μg (1 mg/kg) of each antibody was injected intraperitoneal injection (i.p.) into NOG mice; after 24 hours of antibody administration, the mice were challenged through the i.p. route with 500 TCID50 live HIV-Shenzhen-BC.

Please refer to lines 174-178.

  1. The color coding in Figure 1f should be clearly defined for better understanding.

Response: We have revised Figure 1F according to your suggestion. (F) Illustration of the design of the tri-specific antibody in this study. Blue: variable region of monoclonal antibody 1, yellow: variable region of monoclonal antibody 2, red: variable region of monoclonal antibody 3, light gray: human IgG1 CH1, dark gray: human IgG1 CH2, black: human IgG1 CH3.

  1. Certain result descriptions are inaccurate. For instance, the data in Figure 2 indicate that VRC01 failed to neutralize 4 out of 18 pseudoviruses, not 2 as stated.

Response: Thank you very much for your correction. We have revised the relevant statements.

Reviewer 2 Report

Comments and Suggestions for Authors

This study presents a novel approach/platform for the development of a single potent antibody with trispecific binding sites that broadly neutralize global HIV strains. It is succinct, nicely written, and highly engaging, however, it should be expanded upon as needed. I have some comments/remarks for authors to consider in order to enhance the quality and clarity of their manuscript.

·        The authors may consider emphasizing in the introduction the several forms of pre-exposure prophylaxis (PrEP) that are already available, including daily tablets TDF/3TC and long-acting injectable cabotegravir (CAB-LA) for HIV. The authors should briefly address in their discussion how their novel trispecific antibodies against HIV will address some of the limitations of these WHO-recommended PrEPs that are already on the market if their candidate trispecific antibodies are shown to be safe and effective in randomized clinical studies. This supports the concept behind the development of tri-specific antibodies against HIV as a substitute for the existing PrEP and antiretroviral treatments.

·        Because it impacts the safety of the animal model challenge experiment, information regarding the purity of the recombinant trispecific antibodies following purification by size-exclusion chromatography is crucial. If the authors have this data please submit it as a supplementary.

·        Data on the expression efficiency of tri-specific antibodies are crucial for predicting the influence of production efficiency on these new products' future cost-effectiveness and affordability. Please include the data, if available, from small-scale production (mg/Liter ) of transfected HEK293F cell culture.

·        Do the authors have data on Kd for each binding site of the three trispecific antibodies? If so provide it.

·        Who initially developed the monoclonal antibodies that were used to construct the three trispecific antibodies used in this study? Are these mAbs currently patent-free?

·        Surprisingly, but intriguingly, the three trispecific antibodies developed by the authors demonstrated comparable in vivo antiviral effectiveness as seen in Figure 4, despite their varying neutralizing potencies when tested in vitro. What plausible explanation exists for this observation? and the discussion has to highlight this.

·        The authors of this work reported that in order to create unique trispecific antibodies against HIV, they employed three mAbs that targeted the CD4 receptor, CCRS co-receptor, and important functional epitopes of the HIV-1 virus. For a better understanding, a table summarizing the mAbs with their matching targets—the envelope glycoprotein, the CCR5 co-receptor, and the CD4 binding site—is required.

·        If the authors had to make a decision based just on the results from this study, which of the three trispecific antibodies produced in this investigation would they choose to keep using for further non-human primate challenge experiments? The answer to this should be reflected in the discussion section.

Minor comments

·        Line 111. The text "using the GraphPad Prism 6.0 software by. fit the neutralization dose-response curves" is missing words/phrases after "by."

·        For the HIV strain used in the challenge experiment, HIV-Shenzhen-BC, the accession number is necessary.

·        To enhance understanding, it is necessary to have keys that indicate the various colors of the trispecific antibody constructed in Figure 1 F ((F) Illustration of the design of the tri-specific antibody in this study).

·        The binding and neutralizing activity of a single and tri-specific antibody are displayed in Figure 1. This figure title is incorrect since the binding activity as measured by the ELISA is shown on the Y axis (OD450 values) and does not indicate the neutralizing activity of the single antibody and trispecific antibodies.

·        The y-axis in Figure 3 needs to be labeled with IC50 value (ug/mL).

·        Line 253. “In a previous study, it was found that the combination of iMab and 10E8 bispecific antibody could neutralize almost all the HIV-1 pseudoviruses tested…” Here, a reference is required.

·        Lines 263-266. “The trispecific antibody designed in this study did not show ADCC killing activity on CD4+CCR5+ cells, which may be due to the binding of CH3 of the trispecific antibody with ScFv hampering the target of CH3 to NK cells, indicating that these trispecific antibodies have good safety”. Is there any evidence that supports this hypothesis?

Comments on the Quality of English Language

I have already expressed. The authors need to review carefully to make corrections on some of the typo errors. 

Author Response

Review 2

This study presents a novel approach/platform for the development of a single potent antibody with trispecific binding sites that broadly neutralize global HIV strains. It is succinct, nicely written, and highly engaging, however, it should be expanded upon as needed. I have some comments/remarks for authors to consider in order to enhance the quality and clarity of their manuscript.

Response: Thank you very much for your comments, we have supplemented the experiment according to your suggestions, expanded and improved the paper's content.

  • The authors may consider emphasizing in the introduction the several forms of pre-exposure prophylaxis (PrEP) that are already available, including daily tablets TDF/3TC and long-acting injectable cabotegravir (CAB-LA) for HIV. The authors should briefly address in their discussion how their novel trispecific antibodies against HIV will address some of the limitations of these WHO-recommended PrEPs that are already on the market if their candidate trispecific antibodies are shown to be safe and effective in randomized clinical studies. This supports the concept behind the development of tri-specific antibodies against HIV as a substitute for the existing PrEP and antiretroviral treatments.

Response: We have added the relevant information in the introduction and in the discussion section as you suggested.

Anti-HIV-1 drugs can not only be used for post-viral infection treatment but also for pre-exposure prevention of HIV-1. Currently, the primary pre-exposure prophylactic drugs for HIV-1 that have been marketed are Tenofovir Alafenamide (TAF)[2, 3] and Dapivirine[4, 5]. However, these drugs may have some side effects, and there are also some HIV-1-resistant strains[6, 7]. Developing new and practical pre-exposure prophylactic drugs against HIV-1 can help humans more effectively prevent HIV-1 infection (lines 53-59).

The novel trispecific antibodies against HIV-1 developed in this study will address some of the limitations of the WHO-recommended pre-exposure prophylactic drugs against HIV-1 that are already on the market due to the following reasons[12, 28, 40]: (1) The trispecific-antibody drugs possess a lower risk of drug resistance: in contrast to trispecific antibodies, certain drugs may pose a risk of pathogen development of tolerance to the medication. (2) Trispecific antibodies exhibit an extended half-life, allowing for sustained immune protection over a defined period. (3) Trispecific antibodies target three distinct antigens of HIV; in contrast, certain traditional antiviral drugs may lack the same specificity, resulting in a lesser impact on host cells. (4) Trispecific antibodies have demonstrated the ability to reactivate and eliminate latently infected cells under prolonged suppression by antiretroviral therapy (ART). Please refer to lines 393-403.

  • Because it impacts the safety of the animal model challenge experiment, information regarding the purity of the recombinant trispecific antibodies following purification by size-exclusion chromatography is crucial. If the authors have this data please submit it as a supplementary.

Response: We supplemented the SDS-PAGE data of antibody, and the result showed that the antibody purity was good.

The monoclonal antibodies and their recognition targets involved in this study are shown in Table S1. Through HEK293F expression and Protein A purification, we obtained highly purified monoclonal antibodies iMab, PRO140, 10E8, PGT121, PGDM1400, VRC01, trispecific antibodies iMab+PRO140+10E8, iMab+PRO140+PGMD1400, and iMab+PRO140+PGT121 were expressed and purified (Figure S1). Please refer to lines 194-199.

Figure S1. SDS-PAGE analysis of monoclonal antibodies iMab, PRO140, 10E8, PGT121, PGDM1400, VRC01,  trispecific antibodies iMab+PRO140+10E8, iMab+PRO140+PGMD1400, and iMab+PRO140+PGT121 expression and purification. Lane 1: Marker;  Lane 2. iMab (5 μg); Lane 3. PRO140 (5 μg);  Lane 4.10E8 (5 μg);  Lane 5. PGMD1400 (5 μg);  Lane 6. PGT121 (5 μg);  Lane 7. VRC01 (5 μg);  Lane 8. Trispecific antibody iMab+PRO140+10E8 (1 μg); Lane 9. Trispecific antibody iMab+PRO140+PGMD1400 (1 μg);  Lane 10. iMab+PRO140+PGT121 (1 μg); Lane 11. Trispecific antibody iMab+PRO140+10E8 (5 μg); Lane 12. Trispecific antibody iMab+PRO140+PGMD1400 (5 μg);  Lane 13. iMab+PRO140+PGT121 (5 μg).

  • Data on the expression efficiency of tri-specific antibodies are crucial for predicting the influence of production efficiency on these new products' future cost-effectiveness and affordability. Please include the data, if available, from small-scale production (mg/Liter ) of transfected HEK293F cell culture.

Response: The small-scale production of transfected HEK293F cell culture of the three tri-specific antibodies iMab+PRO140+10E8, iMab+PRO140+PGMD1400, and iMab+PRO140+PGT121 is 5-10 mg/liter (Lines 113-117).

The three trispecific antibodies developed in this study have high small-scale expression efficiency in HEK293F cells, indicating that their production costs can be effectively controlled and they have good affordability as marketed drugs in the future  (Lines 388-391).

  • Do the authors have data on Kd for each binding site of the three trispecific antibodies? If so provide it.

Response: We supplemented the Kd data for the affinity of three tri-specific antibodies to the HIV-1 antigens.  

In addition, binding of trispecific antibody iMab+PRO140+10E8 to HIV-1 gp41 protein, trispecific antibodies iMab+PRO140+PGMD1400 and iMab+PRO140+PGT121 to HIV-1 gp120 protein, were measured by surface plasmon resonance, with the HIV-1 proteins being strongly bound by the trispecific antibodies: Kd (dissociation constant) = 6.12 nM for iMab+PRO140+10E8 versus HIV-1 gp41 protein; Kd = 0.46 nM for iMab+PRO140+PGMD1400 versus HIV-1 gp120; Kd = 0.59 nM for iMab+PRO140+PGT121 versus gp120. (Figure S2). Lines 215-222.

Figure S2. Binding curves for reported Kd values for binding of tree trispecific antibodies to HIV-1 antigens. The other curves are the experimental traces from testing various concentrations of antibodies in SPR experiments. (A) Trispecific antibody iMab+PRO140+10E8 versus gp41. (B) Trispecific antibody iMab+PRO140+PGMD1400 versus gp120. (C) Trispecific antibody iMab+PRO140+PGT121 versus gp120.

  • Who initially developed the monoclonal antibodies that were used to construct the three trispecific antibodies used in this study? Are these mAbs currently patent-free?

Response: The iMab was developed by Tanox Inc (under license from Biogen Idec, formerly Biogen Inc) around the year 2000[35], PRO-140 was developed by Progenics Pharmaceuticals, Inc. in the 1990s[36], 10E8 was developed by the Mark Connors team at the National Institutes of Health in 2012[15], PGT121 was developed by Julie Overbaugh's team from Fred Hutchinson Cancer Research Center in the United States in 2012[37], PGDM1400 was developed by Dennis R Burton's team from the Massachusetts Institute of Technology and Harvard University in 2014[16]. Lines 352-359.

Most of these antibodies have patent protection, however, it does not affect the application of these antibodies in related research. Currently, these antibodies have been studied by multiple research teams worldwide and hundreds of related research papers have been published, the relevant patents are as follows:

10E8:https://www.rainpat.com/Home/Patentinfo?id=0780126C3BFC8B0C2BB69F7DA94B137E&dk=662212CE30D95E97&sid=ccb6d41f-394b-4899-8815-e7c946166bc6&tss=0780126C3BFC8B0CFD3E32A127824EBF6FB705235CDC0B2E10AA2C69B9DB61CB

PGT121:https://www.rainpat.com/Home/Patentinfo?id=0E842415A29E3BF0BB9DB5989A457A08&dk=4BB79C4305D156B5&sid=a7d34f9b-ef0b-48c6-852c-d430cb5672a4&tss=0E842415A29E3BF09371F5DEF5525A24A580C52FE99C60E07A0EF0E015D19D64C58DD5B741C64D91

https://www.rainpat.com/Home/Patentinfo?id=C1F6DD72F559527DC63B7B65F8B4C8F1&dk=4BB79C4305D156B5&sid=a7d34f9b-ef0b-48c6-852c-d430cb5672a4&tss=C1F6DD72F559527D9DF178E5CDE51F04AA2427C9240C4E32F1D2A30D77882740

PGDM1400:

https://www.rainpat.com/Home/Patentinfo?id=C4ED2671861F9F935E3180FB73197E3A&dk=4BB79C4305D156B5&sid=2f7e93ca-28ab-417a-92f0-dd1bb72d7389&tss=C4ED2671861F9F939B19E8E9A26F2DAE1C96961DC58DBB1A0D8A3C765A25F059B5560526B17F61F5

  • Surprisingly, but intriguingly, the three trispecific antibodies developed by the authors demonstrated comparable in vivo antiviral effectiveness as seen in Figure 4, despite their varying neutralizing potencies when tested in vitro. What plausible explanation exists for this observation? and the discussion has to highlight this.

Response: In the revised manuscript, we added a dose group of 1 mg/kg. When the antibody was administered at 1 mg/kg,  HIV-1 virus could be detected on the third day after the infection of the mice in the iMab+PRO140+10E8-treated group, and the virus titer gradually increased during the observation period, but the virus titer of the mice treated by the iMAB +PRO140+10E8 group was significantly reduced compared with the negative antibody control group. Encouragingly, no HIV-1 virus was detected in mice treated with 1 mg/kg of iMab+PRO140+PGMD1400 and iMab+PRO140+PGT121 (Figure 4C).

Please refer to lines 309-316.

  • The authors of this work reported that in order to create unique trispecific antibodies against HIV, they employed three mAbs that targeted the CD4 receptor, CCRS co-receptor, and important functional epitopes of the HIV-1 virus. For a better understanding, a table summarizing the mAbs with their matching targets—the envelope glycoprotein, the CCR5 co-receptor, and the CD4 binding site—is required.

Response: We have supplemented Table S1 according to your suggestion.

Table S1. Summary of monoclonal antibodies and their recognition targets used in this study.

Antibody

Target

iMab

CD4 receptor

PRO140

CCRS co-receptor

10E8

Membrane-proximal external region of gp41 of HIV

PGMD1400

V2 apex of HIV

PGT121

N332 glycan supersite of HIV

VRC01

CD4 binding site of HIV

  • If the authors had to make a decision based just on the results from this study, which of the three trispecific antibodies produced in this investigation would they choose to keep using for further non-human primate challenge experiments? The answer to this should be reflected in the discussion section.

Response: Among the three trispecific antibodies in this study, iMab+PRO140+PGDM1400 showed the best overall neutralizing activity against HIV-1 and demonstrated decent in vivo anti-HIV-1 activity in humanized mouse models. Therefore, in further research, our team will choose the trispecific antibody iMab+PRO140+PGDM1400 to conduct further in vivo assay to determine the antiviral efficacy of this antibody in non-human primates (lines 363-368).

Minor comments

  • Line 111. The text "using the GraphPad Prism 6.0 software by. fit the neutralization dose-response curves" is missing words/phrases after "by."

Response: We have made the necessary modifications as requested. The IC50 values were determined using the GraphPad Prism 6.0 software by fitting the neutralization dose-response curves using nonlinear regression (Lines 152-154).

  • For the HIV strain used in the challenge experiment, HIV-Shenzhen-BC, the accession number is necessary.

Response:  We have added the whole genome sequence of HIV Shenzhen BC in the supplementary materials.

Whole genome sequence of HIV Shenzhen BC strain:

TAGTGGCGCCCGAACAGGGACCTTGAAAGCGAAAGTAGAACCAGAGGAGATCTCTCGACGCAGGACTCGGCTTGCTGAAGTGCACTCGGCAAGAGGCGAGAGCGGCGGCTGGTGAGTACGCCAATTTTATTTGACTAGCGGAGGCTAGAAGGAGAGAGATGGGTGCGAGAGCGTCAGTATTAAGAGGCGGAAAATTAGATAAATGGGAAAAAATTAGGTTAAGGCCAGGGGGAAAGAAAAAATATAGGCTAAAACACCTAGTATGGGCAAGCAGGGAGCTGGAAAAATTTGCACTTAACCCTGACCTTTTAGAGACAGCAGAAGGCTGTAAACAAATAATAAAACAGTTACACCCAGCTCTTAAGACAGGAACAGAAGAACTTAAATCATTATTCAACTTAGTAGCAGTTCTCTATTGTGTACATTCAAATATAGATGTGAAAGACACCAAAGAAGCCTTAGACAAGATAGAGGAAGAACAAAACAAAATTCAGCAAAAAACACAGCAGGCAAAGGAGGCTGACAAAAAGGTCAGTCAAAATTTTCCTATAGTGCAGAATATCCAAGGGCAAATGGTACATCAGCCCATATCACCTAGAACYTTAAATGCATGGGTAAAAGTGGTAGAAGAAAAGGCTTTTAGCCCAGAAGTAATACCCATGTTTTCAGCATTATCAGAAGGAGCCACCCCACAAGATTTAAACACCATGCTAAACACAGTGGGGGGACATCAAGCAGCCATGCAAATATTAAAAGACACCATCAATGAAGAAGCTGCAGAATGGGATAGATTACATCCAGTACATGCAGGGCCTGCTGCACCAGGCCAAATGAGAGAACCAAGGGGAAGTGACATAGCAGGAACTACCAGTACCCTTCAGGAACAAATAGCATGGATGACGAGTACTCCACCTGTTCCAGTAGGAGACATCTATAAAAGGTGGATAATTCTGGGATTAAATAAAATAGTAAGAATGTATAGCCCCACCAGCATTCTGGACATAAAACAAGGGCCAAAGGAACCCTTTAGAGAYTATGTAGACAGGTTCTTTAAAACTTTGAGAGCAGAACAAGCTACACAAGATGTAAAAAATTGGATGACAGACACCTTGTTGGTCCARAATGCGAACCCAGATTGTAAGACCATTTTAAGAGCATTAGGACCAGGGGCTTCAATAGAAGAAATGATGACAGCATGTCAGGGAGTGGGAGGACCTAGCCATAAAGCAAGAGTGTTGGCTGAGGCAATGAGCCAAACAAACAATGCCATAATGATGCAGAGAAGCAATTTTAAAGGTCCTAAAAGAATTATTAAATGTTTCAACTGTGGCAAGGAAGGGCACATAGCCAAAAATTGTAGGGCCCCTAGAAAAAAGGGCTGTTGGAAATGTGGGAAAGAAGGACACCAAATGAAAGACTGTACTAATGAGAGACAGGCTAATTTTTTAGGGAAAATCTGGCCCTCCCACAAGGGAAGGCCAGGGAATTTTCTTCAGAGCAGGCCAGAACCAACAGTCCCAACAGCCCCACCAGAGGAGAGCTTCAGGTTTGGGGAAGAGATGACAACTCCATCTCAAAAGCAGGAGCCAGTAACTTCCCTCAAATCACTCTTTGGCAACGACCCCTTGTTACCATAAGGATAGGGGGGCAATTAAAAGAAGCTCTATTAGATACAGGAGCAGATGATACAGTATTAGAAGAAATGAATTTGCCAGGGAAATGGAAACCAAAAATGATAGGGGGAATTGGAGGTTTTATCAAAGTAAGACAGTATGAAGAGRTACCCATAGAAATCTGTGGACACAAAGTTATAGGTACAGTATTAATAGGACCTACACCTGTTAACATAATTGGAAGAAATCTGTTAACTCAGCTTGGTTGTACTTTAAATTTTCCAATCAGTCCTATTGAAACTGTACCAGTAAGACTAAAGCCAGGAATGGATGGCCCAAAGGTTAAACAATGGCCATTAACAAAAGAGAAAATAGAAGCATTAACAGAAATTTGTAAAGAAATGGAAAAGGAAGGAAAAATTACAAAAATTGGGCCTGAAAATCCATACAACACTCCAATATTTGCTATAAAAAAGAAAGACAGTACTAAGTGGAGAAAATTAGTAGATTTCAGGGAACTCAATAAAAGAACTCAAGATTTTTGGGAGGTTCAATTAGGAATACCACACCCAGCAGGATTAAAAAAGAAAAAATCAGTGACAGTGCTGGATGTGGGGGATGCATATTTTTCAGTTCCTTTAGATGAAGACTTCAGAAAATATACTGCATTCACCATACCTAGTGTAAACAATGRAACACCAGGGATTAGGTATCAGTACAATGTACTTCCACAGGGATGGAAGGGATCACCAGCAATATTTCAAAGYAGCATGACAAAAATCTTAGAGCCTTTTAGAAAACAAAATCCAGACATAGTCATCTATCAATACATGGATGATTTGTATGTAGGATCTGACTTAGAAATAGGACAGCATAGAACAAAAATAGAGGAACTGAGAGAACACTTGTTGAGGTGGGGATTTACCACACCAGACAAGAAACATCAGAAAGAACCTCCATTTCTTTGGATGGGGTATGAACTCCATCCCGACAAATGGACAGTACAGCCTATACATCTGCCAGAACAAGATAGCTGGACTGTCAATGATATACAAAAGTTAGTGGGAAAATTAAACTGGGCAAGTCAGATTTATCCTGGAATTAAAGTAAGGCAACTTTGCAAACTCCTTAGGGGGGCCAAAGCACTAACAGACATAGTACCACTAACTGAAGAAGCAGAATTAGAATTGGCAGAAAACAGGGAAATYCTAAAAGAACCAATACATGGAGCATACTATGACCCATCAAAAGACTTGATAGCTGAGATACAGAAACAGGGGCAGGACCAATGGTCATATCAGATTTATCAAGAACCATTCAAAAATCTAAAAACAGGAAAGTATGCAAAAATGAGGACTGCCCACACTAATGATGTAAGACAATTAACAGAAGCTGTGCAGAAGATAGCCATGGAAAGCATAGTAATATGGGGAAAGACTCCTAAATTTAGACTACCCATCCAAAAGGAGACATGGGAGACATGGTGGACAGACTATTGGCAAGCCACCTGGATTCCTGAGTGGGAATTTGTTAATACCCCTCCCTTAGTAAAATTATGGTACCAGCTGGAAAAAGACCCCATAGCAGGAGCAGAAACTTTCTATGTAGATGGAGCAGCTAATAGGGAAACTAAAATAGGAAAAGCAGGGTATGTTACTGACAGAGGAAGGAAAAAAGTTGCTACTCTAAATGAAACAACAAATCAGAAAACTGAATTGCATGCAATCTGTCTAGCGTTGCAAGATTCAGGATCAGAAGTAAACATAGTAACAGATTCACAGTATGCATTAGGGATCATTCAAGCACAACCAGATAAGAGTGACTCAGAGTTAGTTAACCAAATAATAGAACAATTAATAAAAAAGGAAAGAGTCTACCTGTCATGGGTACCAGCACATAAAGGAATTGGAGGAAATGAACAAGTAGATAAATTAGTAAGTAGTGGTATCAGGAAAGTATTATTTCTAGATGGAATAGATAAAGCTCAAGAAGAGCATGAAAGGTATCACAGCAATTGGAGAGCAATGGCTAGTGATTTTAATCTGCCACCCGTAATAGCAAAAGAAATAGTGGCTAGCTGTGATAAATGTCAGTTAAAAGGAGAAGCCATGCATGGACAAGTAGACTGTAGTCCAGGGATATGGCAATTAGATTGTACCCATYTAGAAGGAAAGATTATCCTGGTAGCAGTCCATGTAGCCAGTGGCTACATAGAAGCAGAGGTTATTCCAGCAGAAACAGGACAAGAGACAGCATACTTTATACTAAAATTAGCAGGAAGATGGCCAGTCAAAGTAATACATACAGACAATGGTAGTAATTTCACCAGTGCTGCAGTTAAGGCAGCCTGTTGGTGGGCAGGCCTCCAACAGGAATTTGGAATTCCCTACAATCCCCAAAGTCAGGGAGTAGTAGAATCCATGAATAAAGAATTAAAGAAAATTATAGGGCAGGTAAGAGATCAAGCTGAACACCTTAAGACAGCAGTACAAATGGCAGTATTCATTCACAATTTTAAAAGAAAAGGGGGGATTGGGGGGTACAGTGCAGGGGAGAGAATAATAGACATAATAGCAACAGACATACAAACTAAAGAATTACAAAAACAAATTACAAAAATTCAAAATTTTCGGGTTTATTWTCAGRGACAGCAGAGACCCCATTTGGAAAGGACCAGCCAAACTACTCTGGAAAGGTGAAGGGGCAGTAGTAATACAAGATAATAGTGACATAAAGGTAGTACCAAGGAGGAAAGCAAAAATCATTAAGGGACTATGGMAAACAGATGGCAGGYGMTGATTGTGTGGCAGGTAGACAGGATGAAGATTAGAACATGGAATAGTTTAGTAAAACACCATATGTATGTTTCCAGAAGAGCTAATGGATGGTTCTACAGACATCATTATGAGAGCAGACATCCGAAAATAAGTTCAGAAGTACACATCCCAATGGGAGAAGCTAAATTAGTAATAAAAACATATTGGGGGTTGCAAACAGGAGAAAGAGATTGGCATTTGGGACATGGAGTTTCCATAGAATGGAGATTCAAAAGATATAGCACACAAATAGAACCTGGTCTAGCAGACCAACTAATTCATTTATATTATTTTGATTGTTTTGCAGACTCTGCTATAAGGAGAGCCATATTAGGACATATAGTAATTCCTAGGTGTGACTATCAAGCAGGGCATAATAAGGTAGGATCTCTACAATATTTGGCACTGACAGCATTGATAAAACCAAAAAGGATAAAGCCACCTCTGCCTAGTGTTAAGAAACTAGTAGAAGATAGATGGAACAAGCCCCAGAAGACCAGGGGCCACAGAGGGAACCATTCAATGGATGGACAATAGAGCTTCTAGAGGAGCTCAAGCAGGAAGCTGTCAGACACTTTCCTAGACCATGGCTTCATGGCTTAGGACAATACATCTATGAAACATATGGGGATACTTGGGCAGGGGTGGAAGCTATAATAAGGATTCTACAACAACTGCTGTTTGTCCATTTCAGAATTGGGTGTCAGCATAGCAGAATAGGCATTTTGAGACAGAGAAGAACAAGAAATGGAGCCAGTAGATCCTAAATTAGAGCCTTGGAAGCATCCAGGAAGTCAGCCTAAGACTGCTTGTACAAATTGCTATTGTAAAAGATGCTGCTTACACTGCCAAGTTTGTTTCATGCAAAAAGGCTTAGGAATCTTCTATGGCAGGAAGAAGCGAAGGCAACGACGAAGACCTCCTCAGAGCAGTGAGGATCATCAAAATTCTATACCAAAGCAGTAAGTAGTAAATGTAATGCAAGCTTTAGCCATTTTTGCAATAGTAGCCTTAATAGTAGTAGGAATAATAGCAATAGTTATATGGACAATAGTACTCATAGAATATAGAAAAATATTAAGACAGAGAAAAATAGATAGGTTAATTGATAGAATAAGAGAAAGAGCAGAAGACAGTGGCAATGAGAGTGACGGGGATCAAGAAGAATTGGCATTTTTGGAAATGGGGCACCTTGCTCCTTGGAATGTTGATGATCTGTAGTGCTGCAGCAAACTTGTGGGTCACAGTCTATTATGGGGTACCTGTATGGAAAGAAGCAACCACAACTTTATTTTGTGCATCAGATGCTAAATCATATGATACAGAGGTACATAATGTCTGGGCTACACATGCCTGTGTACCCACAGACCCTAACCCACAGGAAATAGTCTTGGCAAATGTAACAGAAAATTTTAACATGTGGAAAAATGATATGGTAAATCAGATGCATGAAGATGTAATCAGTTTATGGGATCAAGGCCTAAAGCCATGTGTAAAGTTGACCCCACTCTGTGTCACTTTAAATTGTACAGATGTTAGGAATGGTACAAAGAATACCAGTACAAAGAATACCAGTACTATTAGCAATACTAACAGTACTATGAAAAATTGCTCTTTCAATATAACCACAGTATTAAGAGATAAGAAGCAGCAAGTGTATGCACTTTTTTATAAACTTGATATAGTACCACTTAATGATGATAATTCTAGTGATCCTAATGGTGAGTATAGATTAATAAATTGTAATACCTCAACCATGACGCAAGCCTGTCCAAAGATCTCATTTGATCCAATTCCTATACACTATTGCACTCCTGCTGGTTATGCACTTCTCAAGTGTAATGATGAGAATTTCAATGGAACAGGACCATGCCCTAATGTTAGTTCAGTACAATGTACACATGGGATTAAGCCAGTGGTATCAACTCAACTACTATTAAATGGTAGTCTAGCAGAAGATGAAATAGTAGTTAGATCTGAAAATCTGACAAACAATGTCAAAACAATAATAGTAAACCTTAATAAATCTGTAGAAATTGTATGTACAAGACCTAACAATAATACAAGAAAAAGTATAAGGATAGGACCAGGACAAACATTTTATGCAACAGGAGACATAATAGGAGACATAAGACAAGCACATTGTAACATTAGTGGTTGGCAAGACATGTTACATAATGTAAGTAAAAAATTAGCAAAACTCTACCCAAATAAAACAATAATATTTGAACCAGCCTCAGGAGGGGACTTAGAAATTACCACACATAGCTTTAATTGTAGAGGGGAATTTTTCTATTGCAATACATCAGGCCTGTTTAATAGTACATTCAATGGTACATTTACTTTTAATAATACATCCAATGATACAGGAAAGAATGGGAGCATCACAATCCCATGCAGAATAAAGCAAATTGTAAATATGTGGCAGGAGGTAGGACGAGCAATGTATGCTCCTCCCATTGCAGGAAACATAACATGCAAGTCAAATATCACAGGACTGTTATTAGTACGTGATGGAGGGACTGGTAGAGACTCAAATAATACAGAGAAATTCAGGCCTGGAGGAGGAGATATGAGGGACAATTGGAGAAGTGAATTATATAAATATAAAGTGGTGGAAATTAAGCCATTGGGAATAGCACCCACCCAAGCAAAAAGGAGAGTGGTGGGAAAAGAAAAAAGAGCAGTGGGAATAGGAGCTGTGTTCCTTGGGTTCTTGGGAGTAGCAGGAAGCACTATGGGCGCGGCGGCAATGACGCTGACGGTACAGGCCAGACAATTGCTGTCTGGTATAGTGCAACAGCAAAACAATTTGCTGAGAGCTATAGAGGCGCAACAGCATATGTTGCAACTCACAGTCTGGGGCATTAAACAGCTCCAAACAAGAGTCCTGGCTATAGAAAGATACCTAAAGGATCAACAGCTCCTAGGGATTTGGGGCTGCTCTGGAAAACTCATCTGCACCACTGCAGTACCTTGGAACACCAGTTGGAGTAACAAATCTGAACAGGACATTTGGAATAACCTGACCTGGATGCAATGGGATAAAGAAATTAATAATTACACAAACACAATATACAGTTTGCTTGAAGAGGCGCAGAACCAGCAGGAAAGAAATGAGAAAGATCTATTAGCATTGGACAGGTGGGAAAGTCTATGGAATTGGTTTAACATAACGAATTGGCTGTGGTATATAAAAATATTCATAATGATAGTAGGAGGCTTGATAGGCTTAAGAATAATTTTTGCTGTGCTCTCTATAGTGAATAGAGTTAGGCAGGGATACTCACCCTTGTCATTGCAGACCCTTATCCCGAACCCAGAGGGACCCGACAGGCTCAGAAGAATCGAAGAAGAAGGTGGAGAGCAAGACAGAGACAGATCCATTCGATTAGTGACCGGATTCTTAGCACTTGCCTGGGACGACCTGCGGAACCTGTGCCTCTTCAGCTACCACCGATTGAGAGACTTTATATTGGTGACAGCGAGAGTGGTGGAACTTCTGGGACGGAACAGCCTCAGGGGACTACAGAGGGGGTGGGAAATCCTTAAATATCTAGGAAGCCTTGTGCAGTACTGGGGGCAGGAGCTAAAAAAGAGTGCTGTTAGTCTGCTTGATACCATAGCAATAGTAGTAGCTGAAGGAACAGATAGGATTATAGAAGTAGGACAAAGAATTTGTAGAGCTATCTACAACATACCTAGAAGAATAAGACAGGGCTTTGAAGCAGCTTTGCAATAAAATGGGGGCCAAGTGGTCAAAAGGATGCCCTGCTGTAAGGGAAAGAATGAGACGAACTCAGCCAGCAGCAGATAGGAGGGAAAGAAGGAGACAAACTGAACCAGCAGCAGATGGGGTGGGAGCAGCATCTCGAGACCTGGAAAGACATGGAGCAATAACAAGTAGCAATACAGCAGCTACTAATGCTGATGTTGCATGGTTGGAAGCCCAACAGGAGGAGGAGGAGGAAGTGGGTTTTCCAGTCAGACCTCAGGTACCTTTAAGGCCAATGACTTCAAGGGTAGCTTTTGATCTTAGCTTCTTTTTAAAAGAAAAGGGGGGACTGGATGGGTTAGTTTACTCTAAGAAAAGGCAAGAGATCCTTGATTTGTGGGTCTATCACACACAAGGCTTCTTCCCTGATTGGGACAACTACACACCAGGGCCAGGGACCAGGTTCCCACTGACTTTTGGGTGGTGCTTCAAGCTAGTACCAGTTGACCCAAAGGAAGTAGAAGAGATCACCGCAGGAGAAGACAACTGCTTGCTACACCCTGTGTGCCAGCATGGAATGGAGGATGAGCACGGAGAAGTATTAATATGGAAGTTTGACAGCATGCTAGCACGCAGACACATGGCCCGCGAGCTACATCCGGAGTTTTACAAAGACTGCTGACACAGAAGGGACTTTCCGCGGGGACTTTCCACTGGGGCGTGCTGGGAGGTGTGGTCTGGGCGGGACTGGGAGTGGTCAACCCTCAGATGCTGCATATAAGCAGCTGCTTTTCGCCTGTACTGGGTCTCTCTAGTCAGACCAGATTTGAGCCTGGGAGCTCTCTGGCTGCTAGGGAACCCACTGT

  • To enhance understanding, it is necessary to have keys that indicate the various colors of the trispecific antibody constructed in Figure 1 F ((F) Illustration of the design of the tri-specific antibody in this study).

Response: We have revised Figure 1F according to your suggestion. (F) Illustration of the design of the tri-specific antibody in this study. Blue: variable region of monoclonal antibody 1, yellow: variable region of monoclonal antibody 2, red: variable region of monoclonal antibody 3, light gray: human IgG1 CH1, dark gray: human IgG1 CH2, black: human IgG1 CH3.

  •  

      The binding and neutralizing activity of a single and tri-specific antibody are displayed in Figure 1. This figure title is incorrect since the binding activity as measured by the ELISA is shown on the Y axis (OD450 values) and does not indicate the neutralizing activity of the single antibody and trispecific antibodies.

Response: Thank you for your correction. It has been modified.

  • The y-axis in Figure 3 needs to be labeled with IC50 value (ug/mL).

Response: Thank you for your correction. It has been corrected.

  • Line 253. “In a previous study, it was found that the combination of iMab and 10E8 bispecific antibody could neutralize almost all the HIV-1 pseudoviruses tested…” Here, a reference is required.

Response:  The reference has been added.

  • Lines 263-266. “The trispecific antibody designed in this study did not show ADCC killing activity on CD4+CCR5+ cells, which may be due to the binding of CH3 of the trispecific antibody with ScFv hampering the target of CH3 to NK cells, indicating that these trispecific antibodies have good safety”. Is there any evidence that supports this hypothesis?

Response: We have changed the relevant statement.

The trispecific antibody designed in this study did not show ADCC killing activity on CD4+CCR5+ cells, indicating that thess trispecific antibodies has good safety (Lines 387-389).

Reviewer 3 Report

Comments and Suggestions for Authors

In this manuscript, Liang et al., designed and characterized trispecific antibodies against HIV-1 receptor (CD4 and CCR5) and envelop protein in vitro and in vivo. The authors showed that all three trispecific antibodies showed broad neutralizing activity in vitro and antiviral effect in humanized mouse model. Although the manuscript is interesting, there are several concerns that need to be addressed.

 1.       Figure 1: There is no statement regarding reproducibility. How many times did the author perform this experiment with how many wells in each experiment?

2.       Supplemental materials are missing.

3.       Since there are lots of published papers regarding characterization of trispecific antibody against HIV-1 (Examples are listed below.), the novelty and strength of this manuscript is unclear to me. What is the novelty and strength of this manuscript as compared to previous papers?

 (1)    Nat Commun. 2023 Jun 22;14(1):3719. doi: 10.1038/s41467-023-39265-z. Trispecific antibody targeting HIV-1 and T cells activates and eliminates latently-infected cells in HIV/SHIV infections

(2)    Cell Rep. 2022 Jan 4;38(1):110199. doi: 10.1016/j.celrep.2021.110199. Potent anti-viral activity of a trispecific HIV neutralizing antibody in SHIV-infected monkeys

(3)    Nat Commun. 2018 Feb 28;9(1):877. doi: 10.1038/s41467-018-03335-4. Rational design of a trispecific antibody targeting the HIV-1 Env with elevated anti-viral activity

(4)    J Virol. 2018 Aug 29;92(18):e00384-18. doi: 10.1128/JVI.00384-18. Print 2018 Sep 15. Targeting the HIV-1 Spike and Coreceptor with Bi- and Trispecific Antibodies for Single-Component Broad Inhibition of Entry

(5)    Science. 2017 Oct 6;358(6359):85-90. doi: 10.1126/science.aan8630. Epub 2017 Sep 20. Trispecific broadly neutralizing HIV antibodies mediate potent SHIV protection in macaques

Author Response

Review 3.

In this manuscript, Liang et al., designed and characterized trispecific antibodies against HIV-1 receptor (CD4 and CCR5) and envelop protein in vitro and in vivo. The authors showed that all three trispecific antibodies showed broad neutralizing activity in vitro and antiviral effect in humanized mouse model. Although the manuscript is interesting, there are several concerns that need to be addressed.

Response: Thank you for your recognition. We have revised and improved the paper according to your feedback.

  1. Figure 1: There is no statement regarding reproducibility. How many times did the author perform this experiment with how many wells in each experiment?

Response: Thank you very much for your suggestion. We have added relevant information: Each experiment is repeated three times, with double wells for each experiment (Lines 234-235).

  1. Supplemental materials are missing.

Response: Sorry, we have uploaded the supplementary materials.

  1. Since there are lots of published papers regarding characterization of trispecific antibody against HIV-1 (Examples are listed below.), the novelty and strength of this manuscript is unclear to me. What is the novelty and strength of this manuscript as compared to previous papers?

 (1)    Nat Commun. 2023 Jun 22;14(1):3719. doi: 10.1038/s41467-023-39265-z. Trispecific antibody targeting HIV-1 and T cells activates and eliminates latently-infected cells in HIV/SHIV infections

(2)    Cell Rep. 2022 Jan 4;38(1):110199. doi: 10.1016/j.celrep.2021.110199. Potent anti-viral activity of a trispecific HIV neutralizing antibody in SHIV-infected monkeys

(3)    Nat Commun. 2018 Feb 28;9(1):877. doi: 10.1038/s41467-018-03335-4. Rational design of a trispecific antibody targeting the HIV-1 Env with elevated anti-viral activity

(4)    J Virol. 2018 Aug 29;92(18):e00384-18. doi: 10.1128/JVI.00384-18. Print 2018 Sep 15. Targeting the HIV-1 Spike and Coreceptor with Bi- and Trispecific Antibodies for Single-Component Broad Inhibition of Entry

(5)    Science. 2017 Oct 6;358(6359):85-90. doi: 10.1126/science.aan8630. Epub 2017 Sep 20. Trispecific broadly neutralizing HIV antibodies mediate potent SHIV protection in macaques

Response:

Indeed, so far, many HIV-1 trispecific antibodies have been developed. However, compared to these previous studies, this study has the following innovation and significance:

One of the main reasons why HIV-1 can pose a significant challenge to humans is that HIV-1 is very easy to mutate and has many subtypes, however, no matter how the HIV-1 virus mutates, the virus needs to infect cells through cell receptors, therefore, antibodies that competitively bind cell receptors or co-receptors with HIV-1 are up-and-coming antiviral drugs, examples include anti-CD4 binding site antibodies and CCR5 binding site antibodies. Multispecific antibodies can inhibit viral binding to cell receptors by competitively binding to cell receptors or co-receptors and simultaneously inhibit viral replication by binding to important functional epitopes of the HIV-1 virus, thus having dual advantages. Theoretically, this kind of multispecific antibody has dual advantages, possesses stronger antiviral efficacy and is less likely to be escaped by the virus. However, until now, no trispecific antibodies have been reported that simultaneously target the HIV-1 CD4 receptor, CCR5 co-receptor of the host and the functional epitopes of the virus, and this study supplements the current gap. Our study aims to explore the antiviral activity and breadth of this class of trispecific antibodies, to provide a supplement for the development of novel antiviral drugs for HIV-1.

Please refer to Lines 80-84 and Lines 337-350.

Round 2

Reviewer 1 Report

Comments and Suggestions for Authors

The authors have satisfactorily addressed all of my concerns.

Reviewer 3 Report

Comments and Suggestions for Authors

The authors addressed the concerns by reviewers satisfactorily. I have no further comment.